# Synthesis of Cenospheres from Ash and Their Application

Sandugash K. Tanirbergenova [1], Balaussa K. Dinistanova [1,2], Nurzhamal K. Zhylybayeva [1], Dildara A. Tugelbayeva [1], Gulya M. Moldazhanova [1], Aizat Aitugan [1], Kairat Taju [1] and Meruyert Nazhipkyzy [1,2,3,*]

[1] Institute of Combustion Problems, Almaty 050012, Kazakhstan; sandu2201@mail.ru (S.K.T.); b.dinistanova@gmail.com (B.K.D.); nurzhamaljk@mail.ru (N.K.Z.)
[2] Department of Chemical Physics and Material Science, Faculty Chemistry and Chemical Technology, Al-Farabi Kazakh National University, Almaty 050040, Kazakhstan
[3] Department of Materials Science, Nanotechnology and Engineering Physics, Mining and Metallurgical Institute Named after O.A. Baikonurov, Satbayev University, Almaty 050013, Kazakhstan
* Correspondence: meruert82@mail.ru

**Abstract:** The possibility of improving the strength properties of concrete materials based on ash/slag waste from thermal power plants of Almaty (Kazakhstan) by adjusting their chemical composition is considered. An X-ray phase analysis, scanning electron microscopic (SEM) analysis, infrared analysis (IR), and elemental determination analysis (EDAX) of ash and slag wastes were carried out, and additives to correct their chemical composition were selected. The analysis of the conducted studies shows that the addition of polypropylene fiber leads to an increase in the compressive crack resistance compared to the composition of the mixture in which ash is present. The highest compressive strength in which cenospheres increase in strength characteristics is observed on samples modified with 7% cenospheres. It was found that the strength of the concrete with the addition of cenospheres increased by more than two times in comparison with a sample without additives.

**Keywords:** cenospheres; concrete; ash; microspheres





## 1. Introduction

Coal plays a vital role in the production of electricity throughout the world. According to the World Coal Institute, approximately 41% of the world's electricity is produced by coal-fired thermal power plants, and by 2030 it is expected to increase to 44% [1]. However, in addition to electricity, these power plants simultaneously produce large amounts of ash waste. According to statistics, about 750 million tons are produced annually worldwide, and the growth rate is increasing.

Coal fly ash is reused as cenospheres due to their exceptional properties, such as low bulk density, high thermal resistance, high machinability, and high strength. These properties make them suitable for a wide range of industries [2].

At present, increasing the durability of concrete is an important scientific and practical problem. High-quality concretes with a wide range of functionality can be obtained using complex multicomponent additives and composite binders, including those based on the use of technogenic raw materials in the form of dispersed industrial waste. The most important conditions for ensuring increased demand for the proposed building composites are ensuring a constant increase in their efficiency, strengthening the savings regime, and also, especially relevant to the preservation of human civilization, the solution of environmental issues and the transition to comprehensive methods of resource conservation [3,4].

Previous works have demonstrated that considerable attention from researchers has been attracted to using carbon nanotubes (CNTs) as reinforcing materials (cement paste, cement mortar, and concrete) [5–8]. However, CNTs exhibit an elastic modulus of the order of 1 TPa [9,10] with a range of tensile strength from 11 to 63 GPa [11]. CNTs have

been found to improve mechanical properties, accelerate hydration, reduce shrinkage, and improve fire resistance and durability, hence making cementitious materials more stable. At the same time, other researchers have reported a slight improvement or even deterioration in the elastic modulus [12,13]. The elastic modulus can be affected by various factors, including, but not limited to, different surface treatments [14], different types (MWCNTs and SWCNTs) [15], and manufacturing technologies [12]. These conflicting results are mainly due to the dispersion and strength of the interfacial bond between the carbon nanotubes and the cement matrix. Poor dispersion of CNTs leads to void defects and unreacted pockets, resulting in reduced strength. In addition, despite CNTs having higher bond strengths than higher aspect ratios, the higher aspect ratio is the main reason why individual CNTs bond with each other, hindering their dispersion in different environments. This can lead to premature delamination of CNTs from the cement matrix under loading. Thus, CNT characteristics such as length, diameter, and concentration are important parameters that determine both dispersion quality and mechanical properties [16,17]. Another obstacle to the practical application of CNTs is cost. Therefore, the use of carbon nanotubes as reinforcing materials is not effective.

In foreign countries, ash waste is used in various sectors of the economy. The law regulates the mandatory use of up to 25% of waste in construction. At the current pace of industrial development, the use of reinforcing additives is more popular, which makes it possible to obtain concretes with high physical, mechanical, and operational properties. Construction and ash/slag wastes from cenospheres are in great demand, in contrast to the traditional steel reinforcing mesh [18,19], to enhance the strength characteristics of concrete. Recently, modifying additives have been used in the construction industry in the production of cement slurries, self-leveling floors, masonry mortars, etc.

Currently, 520 million tons of ash and slag waste (ASW) have accumulated in Kazakhstan, in which microspheres have already been collected by flotation. The volume of residual waste is increasing by 12 million tons annually. At the same time, ASW occupies large areas, and the construction of ash and slag dumps requires significant capital expenditures on the part of thermal power plants, which ultimately affect the increase in the cost of electricity and heat production. The cost of maintaining one ton of ASW ranges from KZT 2 to 3.5 thousand, or 5–7% of the cost of electricity and heat generation at a pulverized coal thermal power plant. Investments in the reconstruction of one ash and slag dump can reach up to KZT 5 billion, and the construction of a new one costs KZT 10–20 billion.

Nowadays, many researchers are expanding their knowledge in the field of the application of ash/slag microspheres in the production of various types of concrete [20–22]. It was shown in [23–25] that the addition of ash/slag microspheres can significantly increase packing density, thereby simultaneously increasing the fluidity and strength of the product. Ash/slag microspheres from 0 to 60% and various amounts of water were added to cement paste samples. The results showed that the addition of ash microspheres up to 40% significantly increases the packing density of cement materials [26–28]. With the voids partially filled with microspheres and the same volume of water in the voids vacated, the addition of microspheres makes it possible to increase the thickness of the water film of the cement paste. It has been established that the thickness of the water film plays an important role in the rheology, adhesiveness, and strength of the mortar and is a key parameter that is considered when designing mortars and concrete mixtures [29]. Also, in the works of [30–35], the results of the influence of ash/slag microspheres on the mechanical, functional, and structural properties of cement composites are presented.

Improving the efficiency of construction is not possible without the use of modern technological solutions and new building materials, including the use of polymer microfibers introduced into concrete. It is recommended to use polypropylene fiber in the technology of fiber-reinforced concrete, plaster, mounting mortars, and hydraulic and cellular concrete to reduce the separation of mixtures and increase water resistance, frost resistance, and abrasion resistance. Now, the actual direction of research is the development of building materials with a complex set of operational properties, often practically mutually exclusive.

One of these materials can be considered lightweight architectural concrete, which contains hollow glass composite materials in its composition and is characterized by significant strength indicators [36–43].

The work used cenospheres synthesized from ash/slag based on thermal power plants in Kazakhstan. Ash/slag waste is a valuable raw material to produce building materials, alumina, silica, naturally alloyed iron ore concentrates, aluminosilicate hollow microspheres, and rare earth metals and can be effectively used as a re-renewable raw material in various industries. The main aspect of ash/slag waste disposal is the high melting temperature (1600–1700 °C), which directly depends on the heterogeneous chemical composition [22,37]. In this regard, the technology for obtaining ash/slag cenospheres in a thermal plasma flow is environmentally friendly and relevant.

The aim of the study is to develop a technology for obtaining cenospheres from ash and slag waste at a minimal cost and to study its application in enhancing the strength characteristics of concrete.

The scientific novelty lies in the development of a technology for obtaining lightweight concrete using synthesized cenospheres from ash and slag waste.

## 2. Experimental

### 2.1. Materials and Chemicals

Industrial waste was collected from Almaty (Kazakhstan). The basis of the binder was Portland cement (PC) grade M-200 produced by "Vostok Cement" (Almaty, Kazakhstan) which was used as a raw material.

Polypropylene fiber, ash/slag, and cenospheres were used as fillers. Ash samples were taken from the ash dump of the Almaty CHPP-2.

### 2.2. Obtaining Cenospheres

A gas–plasma plant is an apparatus for producing hollow microspheres from slag waste, the design of which includes the following main elements: a plasma torch, a plasma torch power source, a feeder, a receiving hopper, a water supply pump, a cooling tower, a gas cylinder (argon) or an air compressor (Figure 1). Slag wastes (products of burnt coal) are poured into the feeder, which, under the influence of gas from a cylinder or compressor, enter the combustion chamber of the plasma torch. In plasma, under the influence of gas, hollow microspheres are formed from slag. The compressor or gas cylinder is used to supply the carrier gas through the slag powder feeder to the plasma. The hollow microspheres formed in the plasma from the plasma torch chamber fall through a cyclone into a receiving hopper containing water. Water is needed so that the hollow microspheres float to the surface, and the unreacted particles of slag, due to their mass, settle to the bottom of the bunker. The power source creates plasma in the plasma torch. Since the plasma temperature is high, the plasma torch should be cooled; for this, a pump is provided to supply water, which is supplied to the anode part and the plasma torch chamber. But the water passing through the plasma torch is also heated; a cooling plant is provided for its cooling. That is, the pump supplies water to the plasma torch, and hot water through the plasma torch enters the cooling tower, where it is cooled by ventilation. Then, the cooled water is pumped into the plasma torch. The process becomes cyclic, which makes the installation mobile and economical.

The plasma gas is argon. Plasmatron operation parameters: operating voltage of 30 V and current strength of 350–450 A.

### 2.3. Characterization

Analysis of the morphological structure and elemental analysis of the obtained samples of cenospheres and ash/slag were carried out using a Quantum 3D 200i scanning electron microscope (SEM) (Dual System, Philadelphia, PA, USA) and a TM 4000 Plus microscope (Hitachi, Japan).

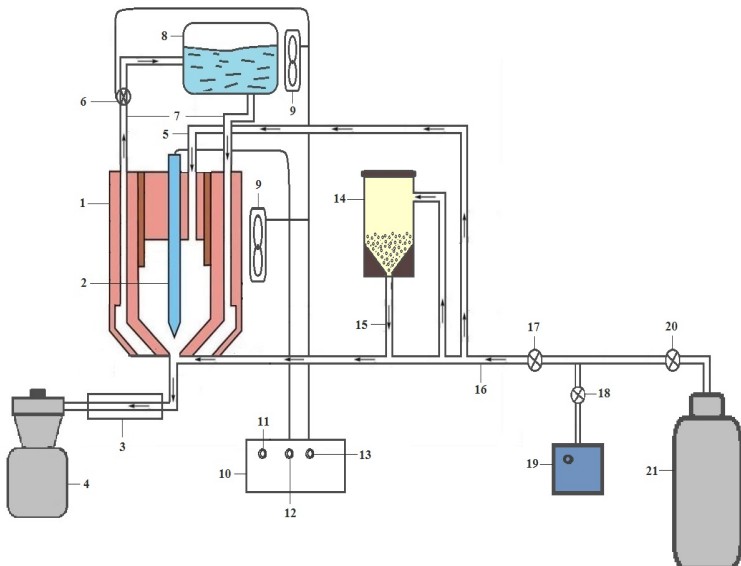

**Figure 1.** Plasmatron operation: 1—plasma torch; 2—electrode (anode); 3—precyclone tube; 4—assembly bunker of the product with a cyclone; 5—plasma gas; 6—pump; 7—cooling water; 8—cooling tower for cooling water; 9—cathode and cooling tower; 10—plasma torch power supply; 11—power supply switch; 12—ignition of the plasma torch; 13—pump switch; 14—feeder (tank for unloading feedstock (ash)); 15—wire for ash/slag supply; 16—gas pipeline; 17—gas supply adjustment valve; 18—compressor shutdown valve; 19—compressor; 20—gearbox with rotameter; and 21—air tank.

The Hitachi TM4000 Plus microscope is equipped with an optional low-vacuum secondary electron detector, which allows you to examine the surface of samples more efficiently with low contrast in back-scattered electrons.

Humidity and water absorption of modified concretes were studied according to the standard method OST-153-39.2-023. Testing of samples to determine the compressive strength (cubes $70 \times 70 \times 70$ mm) was carried out on a small-sized hydraulic press PGM-MG4 according to the standard method. To study the effect of additives on the properties of concrete products, prototypes were made—concrete cubes $70 \times 70 \times 70$ mm in size with the addition of cenospheres and without additives as control samples. To modify the cement, the following were added to the composition of the mixture: polypropylene fiber, synthesized ash/slag waste, and cenosphere in an amount of 3 to 15% by weight of cement. A rotary mixer was used to mix the mixture. The water–cement ratio was 0.25–0.35.

*2.4. Results and Discussions*

Different types of aggregates were considered to improve the strength characteristics of concrete products. The studied ash wastes are a finely dispersed mixture of a predominantly gray color. To assess the quality of the additives used, their main physical and chemical characteristics were studied. The average chemical composition of the studied ash wastes consists of oxide materials with different contents of $SiO_2$ (55.1–58.8%) and $Al_2O_3$ (15.2–21.7%), $Fe_2O_3$ (8.7–10.2%), $MgO$ (1.2–3.6%) and $CaO$ (5.6–10.8%), and amorphous carbon.

The conducted studies have shown that ash/slag waste has a high specific surface area (1580 cm$^2$/g), as it is characterized by a high degree of dispersion, which is confirmed by the results of the granulometric composition. Analysis of the granulometric composition of ash waste showed that 23.5% of particles are up to 140 microns in size, 58% are from 140 to 315 microns, and 18.3% are larger than 315 microns.

The value of the bulk density of ash/slag waste is 0.87 g/cm$^3$, and the bulk density of cenospheres is 0.58 g/cm$^3$.

Figure 2 shows the results of the elemental determination analysis of ash/slag waste.

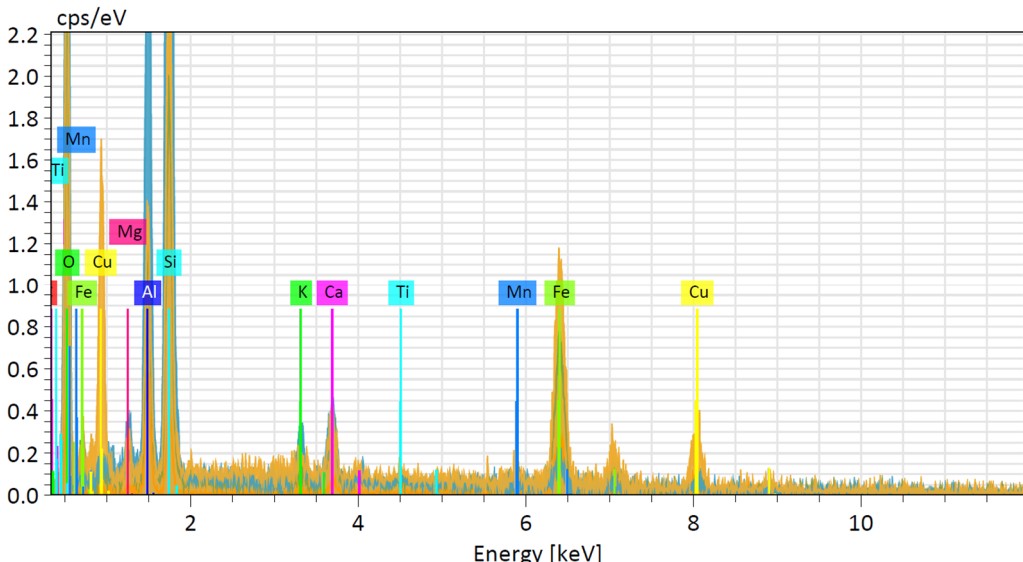

**Figure 2.** Elemental determination analysis of ash/slag waste.

Ash waste consists of up to 98.9% of compounds of ash-forming macroelements (Si, Al, Fe, O, Ca, Ni, Mg, S, K, and Na), and the remaining microelements are contained in a concentration of 0.1% or less.

Figure 3 shows that the particles of ash/slag material are aggregates of various shapes; the particle size ranges from 10 to 630 microns. SEM studies of ash/slag wastes make it possible to visualize the variety of morphological particles in the studied samples.

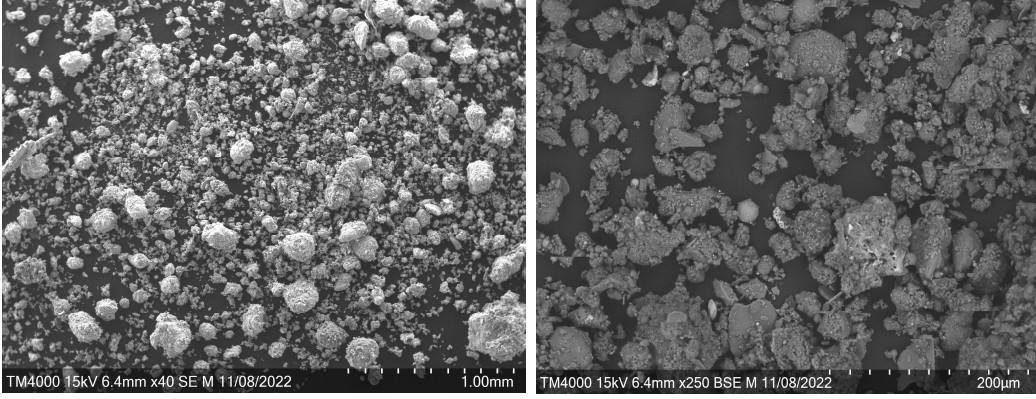

**Figure 3.** Electronic images of ash/slag waste.

Figure 4 shows the energy dispersive X-ray analysis of samples (EDX-analysis) from the surface of the cenospheres, showing that the main elements of the shells of the cenospheres are Al, Si, Fe, K, Na, Mg, Ca, Ti, and O. Traces of copper appear in some samples. A correlation is observed between the phase composition and the Si/Al mass ratio, which is characterized predominantly by the mullite phase $Al_6Si_2O_3$ (Si/Al = 3.5) with an admixture of the X-ray amorphous phase.

The process of formation of the structural-phase composition of the shell of cenospheres based on refractory polycrystalline materials provides the formation of an X-ray amorphous structure of the shell after cooling at a concentration of $SiO_2 \geq 60$ wt.% in the original powder, and the transition of cryptocrystalline modification $\gamma$-$Al_2O_3$ to high-temperature $\alpha$-$Al_2O_3$ with the initial alumina powder.

Figure 5 shows the results of the SEM study of the obtained cenospheres. It was revealed that the cenospheres synthesized from ash waste after plasma treatment have various spherical vitreous forms of carbon aggregates.

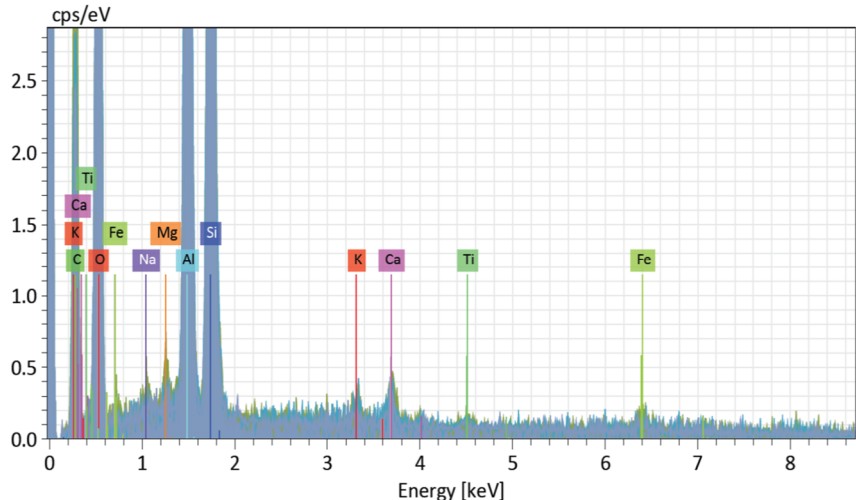

**Figure 4.** Elemental analysis of the synthesized cenospheres.

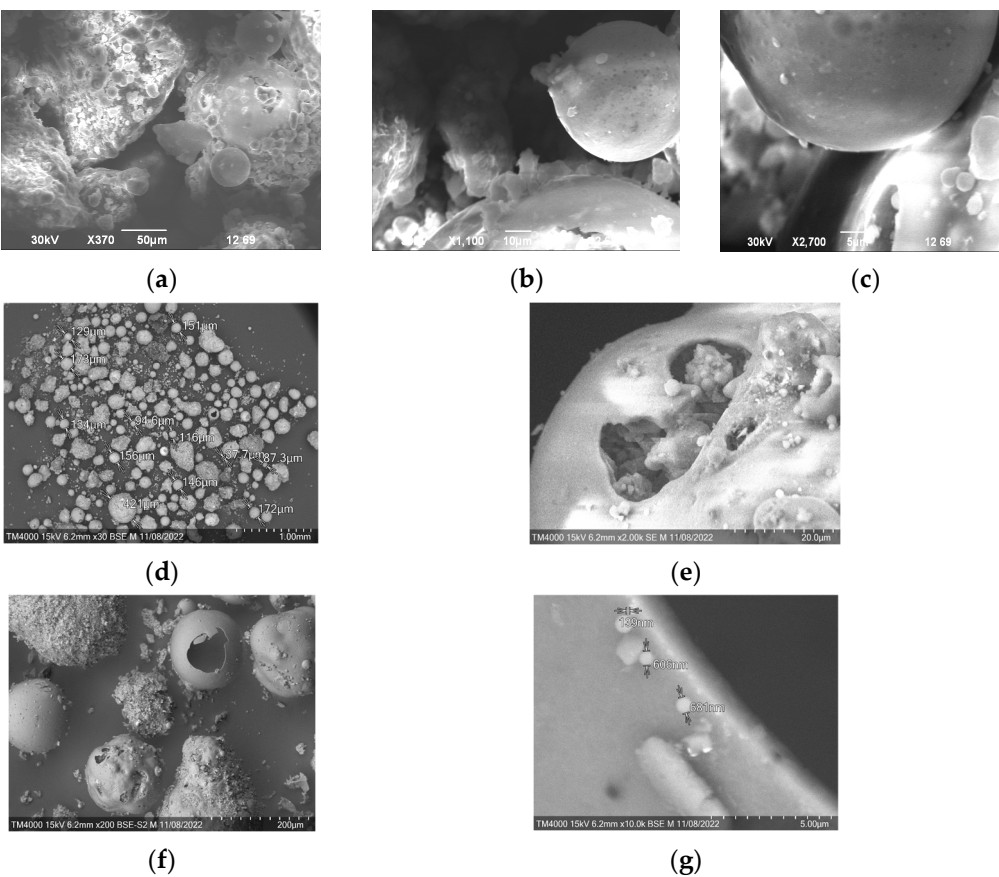

**Figure 5.** Electron microscopic images of cenospheres. (**a**–**c**) some recesses contain the finest particles and accumulated cenospheres of various shapes; (**d**–**g**), the dimensions of the resulting composites range from 130 to 180 µm.

Cenospheres are hollow inside and are black and white free-flowing powders; thin-walled balls with a diameter of 315 microns take the form of cenospheres. The wall thickness of the cenospheres is less than 2 microns, and the cenospheres are solid microballoons. Composites have spherical shapes, provide high strength, chemical resistance, and have low thermal conductivity. It can be seen from the presented images that there are transparent cenospheres covered with a carbon layer. Individual particles have melted craters on the surface, associated with the release of gas on the surface of the particles; such depressions

are mostly melted. Some recesses contain the finest particles and accumulated cenospheres of various shapes (Figure 5a,b). Samples of hollow cenospheres and plerospheres have spherical shapes.

As can be seen from Figure 5d–g, the dimensions of the resulting composites range from 130 to 180 μm. The synthesized cenospheres are characterized by a relative thickness of the shell equal to 0.06 μm, which allows them to be used in the technology of obtaining firing wall materials with a reduced density and thermal conductivity. Based on the SEM analysis, it can be concluded that the size of the cenospheres depends on the size of the ash. An interesting fact is that the amorphous shell is not homogeneous, due to it containing pores (Figure 5), from which it can be concluded that if properly etched, it may form a good retention site for other materials, such as polymers in the creation of polymer-ceramic composites.

Figure 6 shows the results of the X-ray phase analysis of ash waste (a) and the obtained cenospheres (b) and cement (c).

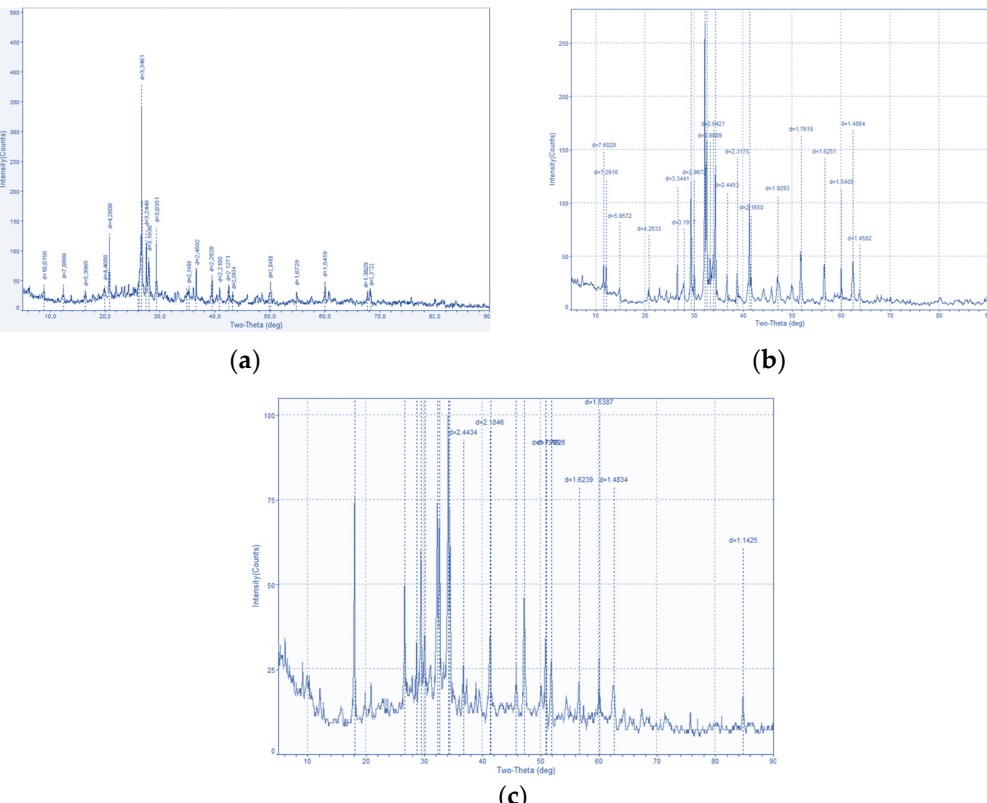

**Figure 6.** X-ray phase analysis: (**a**) ash, (**b**) cenospheres, and (**c**) cenospheres with cement.

The phase compositions of the initial ash wastes and synthesized cenospheres were determined by the XPA method. According to the results of the X-ray diffraction analysis of ash wastes, peaks were identified that are characteristic of mullite ($Al_6Si_2O_3$) and calcite phases. To study phase transitions, an X-ray phase analysis of cenospheres obtained in a plasma flow based on the materials under study was carried out. According to the X-ray phase analysis, the composition of the cenosphere is represented by a mixture of mullite ($Al_6Si_2O_3$) phases and an X-ray amorphous phase (Figure 6).

In the flow of thermal plasma of ash particles, due to high temperature, the structure of cenospheres is formed, which explains the intense bands of specific bands characteristic of various modifications of quartz, and does not affect the position of the intensity and the width of the main absorption bands for all crystalline modifications. From this, we can conclude that the bands common to all modifications of silica, related to stretching ($1100-1200 \text{ cm}^{-1}$) ($700-800 \text{ cm}^{-1}$) and deformation ($450-500 \text{ cm}^{-1}$) vibrations of the

Si–O–Si are due to internal vibrations in silica molecules or ions and are therefore not affected by temperature.

An analysis of the IR spectra of the synthesized cenospheres made it possible to establish that after plasma exposure, the absorption bands related to vibrations of the bonds of low-melting compounds (522.0; 695.0; 779.3; 1163.5 cm$^{-1}$) disappear. The main absorption band at 1092.2 cm$^{-1}$, after heating with the use of plasma, shifts to the long-wave region of 1105.4 cm$^{-1}$, which indicates the beginning of the transition of quartz to tridymite. At temperatures below 870 °C, tridymite can crystallize directly in the metastable form. The inversion between high- and medium-temperature modifications can occur up to a temperature of 125 °C, and between medium- and low-temperature modifications at a temperature of 100 °C.

Ash microspheres from 0 to 10% and various amounts of water were added to cement paste samples. The results showed that the addition of ash microspheres up to 10% significantly increases the density of cement materials. With voids partially filled with microspheres and the same volume of water in voids, the addition of microspheres makes it possible to increase the strength of the cement paste. It has been established that the addition of cenospheres plays an important role in the rheology, adhesiveness, and strength of the mortar, which should be considered when designing mortars and concrete mixes. When the resulting cenospheres are introduced into the concrete mixture, a uniform distribution over the volume and the formation of a homogeneous structure occurs, due to which strength and crack resistance increase, and adhesion to the cement matrix improves.

The characteristics of concretes depend on how strong and durable the bond between the aggregate and the cement slurry is. This parameter is affected by the type of aggregate surface; the roughness can provide good adhesion.

Obviously, the most important factor for the physical and mechanical properties of concrete is the filler. Their content allows you to adjust the density of concrete, as well as its strength, by adjusting the packing density of the particle framework. The strength of concrete will depend on the strength of the cenosphere shell and the strength of the cement–mineral matrix enveloping the filler particles.

Figure 7 shows the results of a study of the effect of ash, cenosphere, and fiber on the compressive strength properties of concrete. The tests carried out showed that the use of hollow carbon–mineral composites (cenospheres) leads to an increase in compressive strength compared to the composition of concrete without additives and with ash.

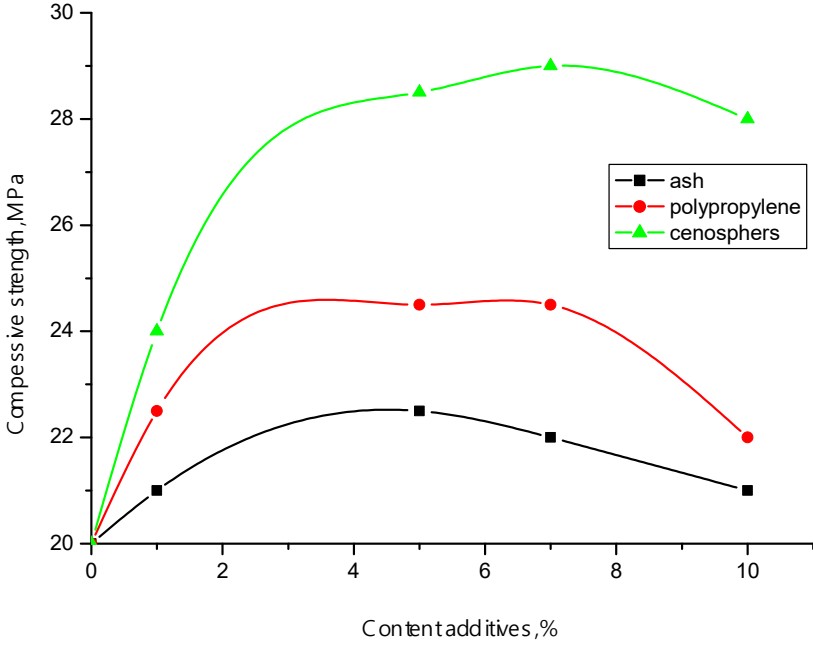

**Figure 7.** Dependence of the influence of additives on the strength of concrete.

The process of the formation of cenospheres is based on agglomerated refractory oxides in a thermal plasma flow. The gas saturation of the hollow microsphere is carried out at the stage of formation of the primary melt film on the surface of the agglomerated particle and is determined by the initial agglomerate porosity. The main expansion of the cenosphere shell (up to 70%) occurs during the heating of the agglomerate to the melting temperature of the condensed phase. With further heating Tmelt < Tvap of a hollow melt drop to the evaporation temperature of the condensed phase, the expansion of the gas cavity does not exceed 7%, which indicates the possibility of controlling the parametric characteristics (diameter and shell thickness) of the obtained hollow spheres.

The analysis of the conducted studies shows that the addition of polypropylene fiber leads to an increase in the compressive crack resistance compared to the composition of the mixture in which ash is present. The highest compressive strength in which cenospheres increase in strength characteristics is observed on samples modified with 7% cenospheres. Reinforcing concrete with 6 mm polypropylene fiber improves strength and reduces weight by 10%. The results of the experiment showed that the compressive strength of concrete products with the addition of cenospheres in comparison with the control is above 29 MPa (Figure 7).

When pure ash is used as an additive, it can be seen from Figure 7 that the concrete strength is low. This is explained by the fact that the ash does not interact with concrete and forms cracks and crevices inside the concrete, thereby reducing strength.

With the introduction of polypropylene into the composition of concrete up to 7%, the concrete crosslinks the polymer molecule, and the compressive strength increases. And with the introduction of more than 7% of polypropylene, a decrease in the fluidity of the concrete mixture is observed, the uniform distribution of the polymer is disturbed, and a decrease in the strength of the concrete is observed.

With the introduction of carbon–mineral cenospheres into the composition of concrete by up to 7%, the hardening and crosslinking of concrete by strong cenospheres occur, and an increase in compressive strength is observed. And with the introduction of more than 7% of the cenospheres, a slight decrease in the fluidity of the concrete mixture is observed, and the uniform distribution of the cenospheres is disturbed; because of this, a decrease in the strength of concrete is observed.

In the cenosphere, consisting of spherical particles with a smooth vitrified surface texture, the mobility of the concrete mixture increases due to a decrease in the internal friction of the concrete mixture. Moreover, the more dispersed the ash, the more vitrified spherical particles in it, and the greater the plasticizing effect it has on the concrete mix.

The influence of ash/slag wastes of different natures on the features of structure formation and properties of concrete is determined; the higher the degree of ash dispersion, the more water is retained on the surface of the particles, and the higher the plasticity of the dispersed phase.

The structure of cenospheres does not change when it is added to concrete, since it underwent maximum heat treatment. The content of mullite and quartzite in the walls of the cenospheres, in the contact zone "cement matrix-microsphere" of calcium hydrosilicates and hydroaluminates, causes high adhesion of the cement stone to the filler. It allows us to obtain high-quality lightweight concretes with specified physical and mechanical properties, which can combine a close-packed structure with a low average density and high strength characteristics.

Experimental studies show it was found that ash waste is a promising raw material base for obtaining cenospheres using gas plasma. Cenospheres with a bulk density of 0.58 g/cm$^3$ were obtained. Numerically, it is shown that the porosity of the initial particle significantly affects the dynamics of heating and can be in various states of aggregation in a plasma flow with a temperature of 1100–1500 °C.

## 3. Conclusions

The conducted studies showed that the best additive that affects the strength of the concrete material based on the ash/slag waste from the Almaty CHPP of Kazakhstan is cenospheres, which consist of spherical particles with a smooth vitrified surface texture. It was found that the strength of the concrete with the addition of cenosphere increased by more than two times in comparison with a sample without additives. The results of the experiment showed that the compressive strength of concrete products with the addition of cenospheres in comparison with the control is above 29 MPa.

Thus, the possibility of obtaining cenospheres based on ash waste in a thermal plasma flow has been established. The phase compositions of ash waste and the morphological structures of the resulting cenospheres have been studied. When using cenospheres, the mass of concrete decreases, but the strength characteristics increase.

**Author Contributions:** Conceptualization, D.A.T. and B.K.D.; validation and formal analysis, G.M.M. and B.K.D.; investigation, A.A. and K.T.; writing—original draft preparation, M.N. and S.K.T.; writing—review and editing, M.N.; visualization, N.K.Z. and S.K.T.; supervision, S.K.T.; project administration, B.K.D. All authors have read and agreed to the published version of the manuscript.

**Funding:** The work was supported by a grant from the Science Committee of the Ministry of Science and Higher Education: AP09260977. "Development of technology for obtaining carbon–mineral composites to enhance the strength characteristics of concrete").

**Data Availability Statement:** All results after we finish project as a report will be in access for everyone by the JSC.

**Acknowledgments:** The authors acknowledge the Science Committee of the Ministry of Science and Higher Education of Kazakhstan for their financial support (Grant No. AP09260977).

**Conflicts of Interest:** The authors declare no conflict of interest.

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
