# Peer review of "Synthesis of Cenospheres from Ash and Their Application"

_jcs, doi:10.3390/jcs7070276_

Round 1

Reviewer 1 Report

Dear Authors,

Your manuscript will require some elaboration. At first the subject has some pre-history and your literature review is rather poor. Try to find more information and summarize the research developments in last 15 years for it. It is well-known subject of research in the start of 2010's. Please indicate clearly the goal, objectives and novelty of your study. Indicate practical application and future tendencies. Abstract and conclusions have to be rewritten.

Fine

Author Response

Response to Reviewer 3 Comments:

According to reviewers comment we revised introduction. We indicated aim of research and its novelty, also it was rewritten the abstract and conclusion.

Reviewer 2 Report

Authors conducted an experimental study to investigate the effects of Cenospheres on the microstructure and compressive strength of cementitious materials. However, there must be a more rigorous interpretation and scientific explanations of the mechanisms of the investigated materials affecting the properties of cement pastes. The manuscript cannot be published in the Journal of Composite Science in its present form. The following comments must be addressed to add value to the manuscript:

General Comment:

1.      The paper should be organized better.

2.      “Experimental program” and “Results and discussions” sections are poorly presented. This must be improved.

Abstract: Please summarize the main points and avoid unnecessary parts. I believe adding some of the most critical quantitative results to the Abstract would be appealing to the readers.

Introduction is somewhat poor (incomplete) given the scope of this study. The most relevant knowledge attained in previous studies should be considered in this manuscript and presented, as a summary. For example, authors stated that The introduction of highly effective additives into concrete makes it possible to obtain a material with improved physical and mechanical properties [1, 2].” Please explain some new additives/materials used in previous studies to improve the properties of concrete and explain their main mechanisms. As an example, recently, nanomaterials such as carbon nanotubes (CNTs) have been used. CNTs were found to increase mechanical properties, accelerate hydration, reduce shrinkage, improve fire resistance, and durability properties, and therefore, make cementitious materials more sustainable. These references might be helpful to discuss: "Probabilistic model for flexural strength of carbon nanotube reinforced cement-based materials", and "Elastic modulus formulation of cementitious materials incorporating carbon nanotubes: Probabilistic approach."

Introduction; Lines 64-66: Authors stated that “Microfiber, when participating in the processes of hydration of cement stone, can partially occupy interstitial spaces and thereby, by reducing the number of holes, control the movement of free water in concrete.” Please discuss microfibers vs. nanofibers/tubes and their mechanisms. Being different from microfibers, nanofibers and nanotubes were reported to prevent or delay the nucleation of cracks at the nanoscale. Such small cracks would neither be regarded as damage nor affect the permeability. This reference might be used to discuss: "Design and Predicting performance of carbon nanotube reinforced cementitious materials: mechanical properties and dispersion characteristics."

Introduction; Lines 75-78: “It has been established that the smaller the diameter of the fibers, the greater their volume content, the greater the surface of their contact with the concrete matrix, which contributes to maintaining the greater internal strength of concrete [25].” Again, nanofibers/tubes might be superior options due to their smaller sizes. The effect of CNT content and physical properties on the pore structure of cementitious materials was discussed in previous research. See these: "Mechanical properties of carbon-nanotube-reinforced cementitious materials: database and statistical analysis", and "Carbon nanotube reinforced cementitious composites: A comprehensive review."

Experimental: This section should be organized better. For example, Section 2.1 does not cover all the raw materials used, and some of them are discussed in Section 2.2; see lines 126-127. Also, lines 128-136 discuss “sample preparation and testing methods” which should be discussed in Section 2.3! In addition, please discuss mix proportion and test matrix.

Results and discussions; Line 184: “Figure 4 shows the results of the SEM study of the obtained cenospheres.” Please replace “Figure 4” with “Figure 5.” Also, correct this in lines 194-195.

Figure 6 & Figure 7: The image quality must be improved.

Results and discussions; Lines 270-271: “Figure 7 shows the results of a study of the effect of ash, cenosphere and fiber on the physical and mechanical properties of concrete.” This figure only discusses the compressive strength, not physical and mechanical properties! Is it 28-day compressive strength? please mention. Also, please discuss the trends and mechanisms. For each material, the compressive strength increased up to certain contents, beyond which it started to drop. This might be due to dispersion issues which result in increased porosity. These references might be used to explain the mechanisms: "Modeling the mechanical properties of cementitious materials containing CNTs", and "Influence of carbon nanotubes on properties of cement mortars subjected to alkali-silica reaction."

Results and discussions; Lines 274-282: Why is this discussed here?

Conclusions must be rewritten to discuss the main findings of this study, not general knowledge.

Moderate English editing is needed. Some of the sentences are grammatically wrong.

Author Response

Response to Reviewer 2 Comments

The following comments must be addressed to add value to the manuscript:

General Comment:

  1. The paper should be organized better.
  2. “Experimental part” and “Results and discussions” sections are poorly presented. This must be improved.

        Response: We agree and did corrections.

Abstract: Please summarize the main points and avoid unnecessary parts. I believe adding some of the most critical quantitative results to the Abstract would be appealing to the readers.

  Response: We agree and revised abstract.

Introduction is somewhat poor (incomplete) given the scope of this study. The most relevant knowledge attained in previous studies should be considered in this manuscript and presented, as a summary. For example, authors stated that The introduction of highly effective additives into concrete makes it possible to obtain a material with improved physical and mechanical properties [1, 2].” Please explain some new additives/materials used in previous studies to improve the properties of concrete and explain their main mechanisms. As an example, recently, nanomaterials such as carbon nanotubes (CNTs) have been used. CNTs were found to increase mechanical properties, accelerate hydration, reduce shrinkage, improve fire resistance, and durability properties, and therefore, make cementitious materials more sustainable. These references might be helpful to discuss: "Probabilistic model for flexural strength of carbon nanotube reinforced cement-based materials", and "Elastic modulus formulation of cementitious materials incorporating carbon nanotubes: Probabilistic approach."

Response: We agree and did corrections.

Introduction; Lines 64-66: Authors stated that “Microfiber, when participating in the processes of hydration of cement stone, can partially occupy interstitial spaces and thereby, by reducing the number of holes, control the movement of free water in concrete.” Please discuss microfibers vs. nanofibers/tubes and their mechanisms. Being different from microfibers, nanofibers and nanotubes were reported to prevent or delay the nucleation of cracks at the nanoscale. Such small cracks would neither be regarded as damage nor affect the permeability. This reference might be used to discuss: "Design and Predicting performance of carbon nanotube reinforced cementitious materials: mechanical properties and dispersion characteristics."

Response: We have revised Introduction. Unnecessary sentences were deleted.

Introduction; Lines 75-78: “It has been established that the smaller the diameter of the fibers, the greater their volume content, the greater the surface of their contact with the concrete matrix, which contributes to maintaining the greater internal strength of concrete [25].” Again, nanofibers/tubes might be superior options due to their smaller sizes. The effect of CNT content and physical properties on the pore structure of cementitious materials was discussed in previous research. See these: "Mechanical properties of carbon-nanotube-reinforced cementitious materials: database and statistical analysis", and "Carbon nanotube reinforced cementitious composites: A comprehensive review."

Response: We have discussed influence of CNT, as below:

Previous works have demonstrated that, considerable attention of researchers has been attracted using carbon nanotubes (CNTs) as reinforcing materials (cement paste, cement mortar, concrete) [5–8]. Although CNTs exhibit an elastic modulus of the order of 1 TPa [9, 10] with a range of tensile strength from 11 to 63 GPa [11]. CNTs have been found to improve mechanical properties, accelerate hydration, reduce shrinkage, improve fire resistance and durability, and hence make cementitious materials more stable. While other researchers have reported a slight improvement or even deterioration in the elastic modulus [12, 13]. The elastic modulus can be affected by various factors, including, but not limited to different surface treatment [14], different types (MWCNTs and SWCNTs) [15] and manufacturing technologies [12]. These conflicting results are mainly due to the dispersion and strength of the interfacial bond between the carbon nanotubes and the cement matrix. Poor dispersion of CNTs leads to void defects and unreacted pockets, resulting in reduced strength. In addition, despite higher bond strengths than higher aspect ratio CNTs, the higher aspect ratio is the main reason why individual CNTs bond with each other, which hinders their dispersion in different environments. This can lead to premature delamination of CNTs from the cement matrix under loading. Thus, CNT characteristics such as length, diameter, and concentration are important parameters that determine both dispersion quality and mechanical properties [16-17]. Another obstacle to the practical application of CNTs is cost. Therefore, the use of carbon nanotubes as reinforcing materials is not effective.

 Experimental: This section should be organized better. For example, Section 2.1 does not cover all the raw materials used, and some of them are discussed in Section 2.2; see lines 126-127. Also, lines 128-136 discuss “sample preparation and testing methods” which should be discussed in Section 2.3! In addition, please discuss mix proportion and test matrix.

Response: According to reviewers comments we did changes in Experimental part.

Results and discussions; Line 184: “Figure 4 shows the results of the SEM study of the obtained cenospheres.” Please replace “Figure 4” with “Figure 5.” Also, correct this in lines 194-195.

Response: Yes, it was technical error. We have corrected.

Figure 6 & Figure 7: The image quality must be improved.

Response: Yes. It was improved.

Results and discussions; Lines 270-271: “Figure 7 shows the results of a study of the effect of ash, cenosphere and fiber on the physical and mechanical properties of concrete.” This figure only discusses the compressive strength, not physical and mechanical properties! Is it 28-day compressive strength? please mention. Also, please discuss the trends and mechanisms. For each material, the compressive strength increased up to certain contents, beyond which it started to drop. This might be due to dispersion issues which result in increased porosity. These references might be used to explain the mechanisms: "Modeling the mechanical properties of cementitious materials containing CNTs", and "Influence of carbon nanotubes on properties of cement mortars subjected to alkali-silica reaction."

Response: Yes. Figure 7 discusses  only compressive strength. It was added discussion, as below:

The resulting cement pastes are molded directly into 70×70×70 cubic steel molds. Then, the molds filled with cement paste are subjected to vibration on a mechanical vibrator for 2 minutes to completely remove air voids and bubbles. Forms are kept for 24 hours at room temperature (22±1 0C). The hardening of concrete is natural, the samples were stored in laboratory conditions at a temperature of 22÷24 0С. The prototypes thus formed are then removed from the molds and transferred to water for compressive strength testing after soaking in water for 28 days.

The formed samples were tested for strength by compression after holding for 28 days. The form was fixed on a vibrating platform, filled approximately 1 cm in height with a solution, and the vibrating platform was turned on. Within 2 minutes of vibration, the mold cavity was evenly filled with mortar. After manufacturing, the samples in the molds were stored for 24 hours in a bath with a hydraulic seal providing a relative humidity of 90%. After that, the samples were disassembled and placed in the baths in a horizontal position. Water should cover the samples by at least 3 cm. The water was changed after 3 days. After a storage period of 28 days, the samples were removed from the water and subjected to a compression test. The compressive strength was calculated as the arithmetic mean of the tests of three samples. Testing was carried out with a Josef Freunol mechanical measuring machine.

Results nd discussions; Lines 274-282: Why is this discussed here?

Response:  We agree with reviewer. Unnecessary part was removed from manuscript.

Conclusions must be rewritten to discuss the main findings of this study, not general knowledge.

Response:  It was rewritten according to the results of study.

Reviewer 3 Report

The problem studied in this manuscript is of the great scientific and technical importance for the researchers, but it needs some revisions and improvement to make it able to be published in this journal

Overall English is good

Author Response

Response to Reviewer 1 Comments

Point 1. The heading is not uniform. The first letter of “Synthesis and Cenospheres” is capital while first letters of other words are small, uniformity should be maintained.

Response 1: Thank you for your comments. We agree and we have corrected.

Point 2. The abstract should be improved. It is confusing.

Response 2: We agree and we revised abstract.

Point 3.  The introduction should be improved and need to be discussed wisely. The literature included should be improved by inserting some of the following articles for improvement:

Colangelo, F., Petrillo, A., & Farina, I. (2021). Comparative environmental evaluation of recycled aggregates from construction and demolition wastes in Italy. Science of The Total Environment798, 149250.

Ferraro, A., Farina, I., Race, M., Colangelo, F., Cioffi, R., & Fabbricino, M. (2019). Pre-treatments of MSWI fly-ashes: a comprehensive review to determine optimal conditions for their reuse and/or environmentally sustainable disposal. Reviews in Environmental Science and Bio/Technology18(3), 453-471.

Ferraro, A., Ducman, V., Colangelo, F., Korat, L., Spasiano, D., & Farina, I. (2023). Production and characterization of lightweight aggregates from municipal solid waste incineration fly-ash through single-and double-step pelletization process. Journal of Cleaner Production, 383, 135275

     Response 3: We agree. We have revised introduction and added new literature and also added 2 literature from reviewer suggestion.

 Point 4.  Figure 6 quality should be improved.

Response 4: We agree. We have redrawn Fig.6.

Round 2

Reviewer 1 Report

No further remarks

Minor

Reviewer 2 Report

Authors have satisfactorily addressed the reviewers' comments.

Reviewer 3 Report

In this manuscript the author discusses about the lightweight concrete from waste materials and used cenospheres as fillers. The author studied the experimentally the concrete behavior with other additives in the compression. The morphological study confirms that cenospheres has spherical hollow shape. With the introduction of the cenospheres to the concrete improve the strength and crack resistance of the concrete.

The problem studied in this manuscript is of the great scientific and technical importance for the researchers, but it needs some revisions and improvement to make it able to be published in this journal.

-      The heading is not uniform. The first letter of “Synthesis and Cenospheres” is capital while first letters of other words are small, uniformity should be maintained.

-      The abstract should be improved. It is confusing.

-      The introduction should be improved and need to be discussed wisely. The literature included should be improved by inserting some of the following articles for improvement about further uses of fly ash in other applications, see for example:

Colangelo, F., Petrillo, A., & Farina, I. (2021). Comparative environmental evaluation of recycled aggregates from construction and demolition wastes in Italy. Science of The Total Environment798, 149250.

Ferraro, A., Ducman, V., Colangelo, F., Korat, L., Spasiano, D., & Farina, I. (2023). Production and characterization of lightweight aggregates from municipal solid waste incineration fly-ash through single-and double-step pelletization process. Journal of Cleaner Production, 383, 135275

-      Figure 6 quality should be improved.